# Promoting the Growth of *Haematococcus lacustris* under High Light Intensity through the Combination of Light/Dark Cycle and Light Color

**Lu Liu** [1,2], **Kezhen Ying** [2], **Kebi Wu** [1,2], **Si Tang** [2], **Jin Zhou** [2] and **Zhonghua Cai** [1,2,*]

1 School of Life Sciences, Tsinghua University, Beijing 100086, China; liu-l19@mails.tsinghua.edu.cn (L.L.); wkb19@mails.tsinghua.edu.cn (K.W.)
2 Shenzhen Public Platform for Screening and Application of Marine Microbial Resources, Shenzhen International Graduate School, Tsinghua University, Shenzhen 518055, China; kezhen.y@outlook.com (K.Y.); si.tang@sz.tsinghua.edu.cn (S.T.); zhou.jin@sz.tsinghua.edu.cn (J.Z.)
* Correspondence: caizh@sz.tsinghua.edu.cn

**Abstract:** The unicellular microalgae *Haematococcus lacustris* is an astaxanthin-rich organism that is widely used for commercial cultivation, but its main limitation is its relatively low biomass yield. It is widely accepted that the use of appropriate high light intensity could promote algal growth; however, *H. lacustris* is very sensitive to high-intensity light, and its growth can be readily arrested by inappropriate illumination. To exploit the growth-promoting benefit of higher light intensities while avoiding growth arrestment, we examined the growth of *H. lacustris* under high light intensities using various light profiles, including different light colors and light/dark cycles. The results show that light color treatments could not alleviate cellular stress under high light intensities; however, it was interesting to find that red light was favored the most by cells out of all the colors. In terms of the light/dark cycle, the 2/2 h light/dark cycle treatment was shown to lead to the highest specific growth rate, which was 46% higher than that achieved in the control treatment (18 μmol/m$^2$·s light intensity, white light). Therefore, in further experiments, the 2/2 h light/dark cycle with red-light treatment was examined. The results show that this combination enabled a significantly higher specific growth rate, which was 66.5% higher than that achieved in the control treatment (18 μmol/m$^2$·s light intensity, white light). These findings offer new strategies for the optimization of illumination for the growth of *H. lacustris* and lay the foundations for more reasonable lighting utilization for the cultivation of commercially valuable algal species.

**Keywords:** *Haematococcus lacustris*; light color; light/dark cycle; light color; high light intensity





## 1. Introduction

*Haematococcus lacustris*, a green algae species, is known for its use in the large-scale industrial production of natural astaxanthin, which is considered to be a product with high economic value and is widely applied in the medicine, food, beauty, and aquaculture industries [1,2]. The cultivation of *H. lacustris* consists of two major phases: the green stage and the red stage [3]. In the green stage, cells are fast-dividing, free-swimming, green biflagellate microalga. When exposed to various stress conditions, *H. lacustris* stop dividing and lose their flagella, and they assemble to become a non-motile, thick-walled, aplanospore-accumulating astaxanthin [4]. Therefore, the cell concentration enrichment in the green stage is the first deciding factor for the astaxanthin production that follows in the red stage [5–7]. With higher cell concentration in the green stage, astaxanthin production could be increased in the red stage. Numerous attempts have been made to achieve higher growth rates in the green stage, such as through the application of light [5]; pH [8,9]; temperature [10,11]; salt concentration [12]; nutrition [13]; and various plant hormones or their derivatives [14].

In this study, we aimed to achieve the faster growth of green-stage cells. Empirically, the growth rate of green-stage cells increases with the increasing light intensity, as described in Equation (1).

$$\mu = \frac{\mu_{max} \times I_{av}^n}{Ik^n + I_{av}^n},\tag{1}$$

where $\mu$ is the specific growth rate, $Ik$ is the constant saturation of irradiance, $I_{av}$ is the average irradiance on the culture surface, $n$ is a shape parameter, and $\mu_{max}$ is the maximum specific growth rate in the culture conditions [15].

However, *H. lacustris* is very light-sensitive, e.g., cells stop division under 60 $\mu$mol/m$^2$·s, which is favored by other algal species [16]. More specifically, for *H. lacustris,* the amount of light energy absorbed by light-harvesting systems may easily exceed the energy consumption limit, causing photoinhibition. The excess energy can produce reactive oxygen species (ROS), resulting in damage to photosynthetic components [17], leading to growth repression [18]. To alleviate these adverse effects [16,19,20] while exploiting the growth-promoting advantages of high light intensity, the employment of different light colors and light/dark cycles needs to be considered.

Different light colors are commonly used in algal cultivation. Primarily, two light colors are used, i.e., blue light (400–500 nm wavelength) and red light (600–700 nm wavelength) [21–25]. It was found that different algal species favor different light colors, for example, *Botryococcus braunii* favors red light [22], whereas *Nannochloropsis sp* favors blue light [26]. For *H. lacustris*, previous light color studies mainly focused on astaxanthin production [27,28]. Only a few articles mentioned that red light could improve the specific growth rate of green-stage cells under low light intensity [29]. To test the potential of light color to alleviate high light stress, blue- and red-light experimental groups were chosen.

For many years, it has been known that photosynthesis is enhanced by light/dark cycles [30,31]. Higher photosynthetic efficiency was found when microalgae were subjected to light/dark cycle treatments compared with constant light exposure. For example, in the mass cultivation of *Nannochloropsis*, an extremely high light condition of 350 $\mu$mol/m$^2$·s was used in the presence of light/dark cycles of 11/11 ms [32]. Regarding *H. lacustris*, Domínguez achieved a higher production rate of vegetative green cells under a light/dark cycle of 12/12 h under 240 $\mu$mol/m$^2$·s with a $CO_2$ supply [33]. However, how *H. lacustris* responds to various light/dark cycles under high light intensity exposure remains unknown, which is one of the focuses of this work.

In this study, we explored whether light/dark cycles and light colors promote the growth of green-stage *H. lacustris* cells under high light intensity. We focused on cellular growth rates under different treatments. Meanwhile, the astaxanthin concentration was used to indicate a cellular stress state, where the emergence of astaxanthin indicated that the environment was not favorable. The intracellular ROS was also monitored. We aimed to identify the best combination of light/dark cycle and light color for *H. lacustris* which could promote cellular growth under the high light regime.

## 2. Materials and Methods

### 2.1. Strain and Cultural Conditions

The strain used in this study was *H. lacustris* FACHB-872, obtained from the Freshwater Algae Culture Collection of Hydrobiology, Chinese Academy of Sciences, China (Wuhan, Hubei Province). Microalgae cells were cultured in Bold's Basal Medium (BBM) in 250 mL Erlenmeyer flasks at 25 °C under a 12/12 h light/dark cycle with the light intensity of 18 $\mu$mol/m$^2$·s. The flasks were shaken manually six times a day.

### 2.2. Experimental Setup

The light/dark cycle and light color experiments were carried out simultaneously. The cells were inoculated in 250 mL sterilized Erlenmeyer flasks which contained 20 mL of seed culture and 80 mL of BBM medium. The starting cell concentration was around $5 \times 10^4$ cells/mL to ensure cells had time to adapt to the high light intensity.

Figure 1 shows the schematic diagram of the experimental setup. To evaluate the influence of different light colors on *H. lacustris* growth, three types of light were included. These were the red-light group: LEDs with a wavelength of 600–700 nm; the blue-light group: LEDs with a wavelength of 400–500 nm; the white-light group: LEDs with a wavelength of 400–700 nm. These groups were subjected to different light intensities, including 18 μmol/m²·s (LL), 55 μmol/m²·s (ML), and 110 μmol/m²·s (HL), for 9 days. The continuous white LED, 18 μmol/m²·s, which is typically used for *H. lacustris* cultivation, was used as the control. In the light color treatments, all of the experiments with all of the groups were carried out under continuous light exposure.

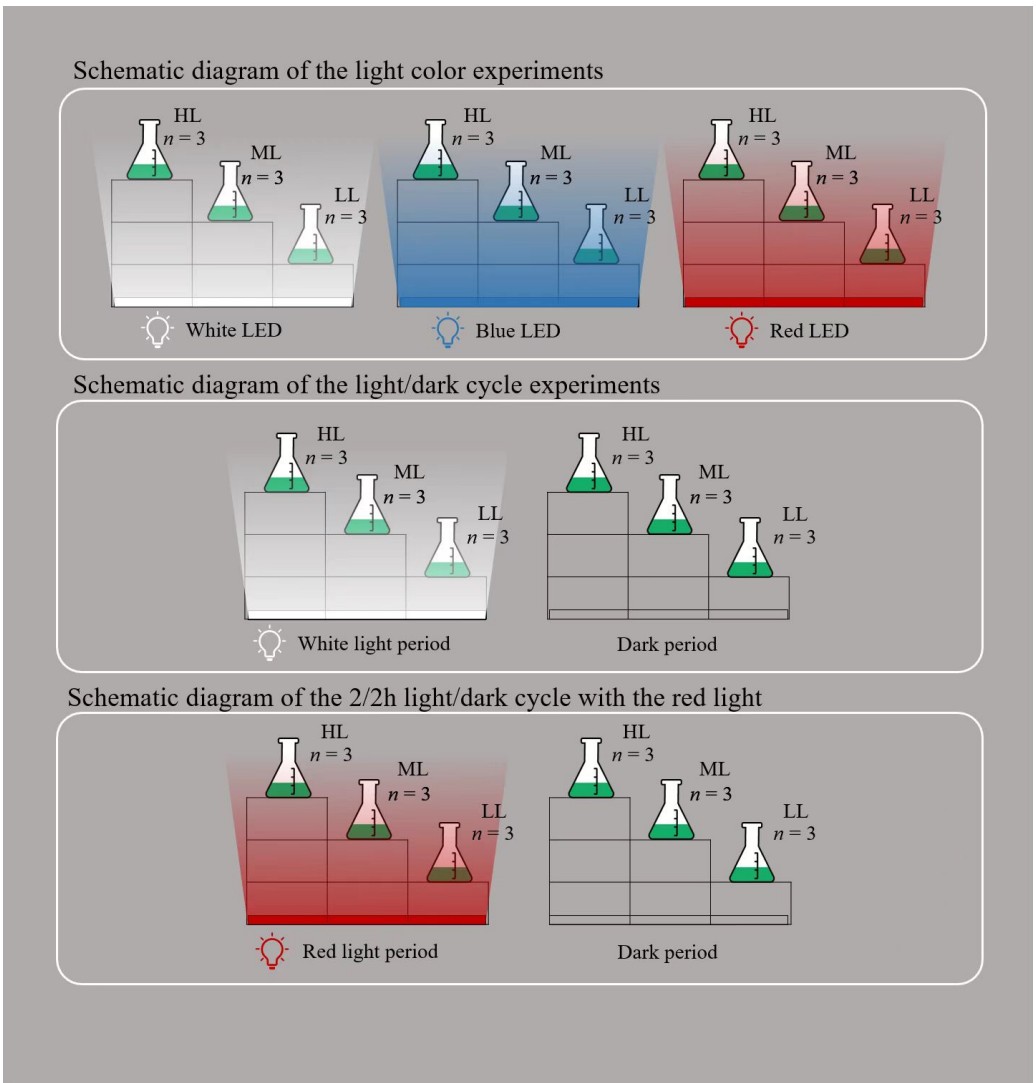

**Figure 1.** Schematic diagram of the experimental setup.

To evaluate the influence of different light/dark cycles, four treatments were established. The 2/2 s group: a two-second dark period and two-second light period with white light; the 2/2 h group: a two-hour dark period and two-hour light period with white light; the 6/6 h group: a six-hour dark period and six-hour light period with white light; the 24/0 h group: the use of white LEDs with continuous light exposure. These experimental groups were subjected to white light under different light intensities, including 18 μmol/m²·s (LL), 55 μmol/m²·s (ML), and 110 μmol/m²·s (HL), for 9 days. The continuous white light exposure group, 18 μmol/m²·s, was used as the control treatment.

Based on the results from the treatments mentioned above, the 2/2 h light/dark cycle with red-light treatment was used to promote *H. lacustris* growth in the green stage. The

cells were inoculated in 250 mL sterilized Erlenmeyer flasks which contained 20 mL of seed culture and 80 mL of BBM medium. In the first experiment, cells could quickly adapt to the higher light intensity. Therefore, to gain more useful data, in the new experiment, the cell concentration was set to $2 \times 10^4$ cells/mL, which is used quite frequently when culturing *H. lacustris*. The cells were subjected to 13 days of treatment to make the final cell concentration more similar to the first experiment. Red-2h-LL: the 2/2 h light/dark cycle with red light under 18 μmol/m$^2$·s; Red-2h-ML: the 2/2 h light/dark cycle with red light under 55 μmol/m$^2$·s. A new control was used (White-24h-LL): continuous white light with a light intensity of 18 μmol/m$^2$·s.

To reduce light occlusion, LEDs were placed under the Erlenmeyer flasks. Different light intensities were obtained by adjusting the heights of the scaffolds. Treatments for all of the groups were repeated in triplicate, and a mean value was reported. The cell concentration, astaxanthin concentration, ROS, and photosynthesis efficiency (Fv/Fm value) were measured every two days during the whole experimental period.

### 2.3. Analytical Methods

The cell concentration and specific growth rate were analyzed. The cell concentration was monitored using a microplate reader, which determined the fluorescence value of the samples [34]. The cell concentration was calculated using the standard curve method, which was used to avoid systematic errors. The optimal excitation and emission wavelengths were 480 nm and 690 nm, respectively. Samples 200 μL in size were placed in a 96-well black plate, and the fluorescence intensity (EX 480, EM690, Gain100) was determined with a microplate reader. The standard equation can express a linear correlation between standard fluorescence intensity and standard samples. The cell concentration was calculated using Equation (2). The specific growth rate is an important parameter that represents the dynamic behavior of microalgae [35]. It could reduce the effect of cell concentration in the growth rate calculation and show the long-term cultivation trend. The specific growth rate was calculated using Equation (3).

$$y = 25.0328 \times (R^2 = 0.914), \tag{2}$$

$$\mu = (\ln M - \ln M_0)/t, \tag{3}$$

where $\mu$ is the specific growth rate, $M_n$ is the final cell concentration, $M_0$ is the initial cell concentration, and t is the time required for the increase in concentration from $M_0$ to $M_n$.

The photosynthetic efficiency ($F_v/F_m$ value) was also measured. The maximum photochemical efficiency of photosystem II (PS II) Fv/Fm, was determined with PHYTO-PAM (Walz, Germany) [36]. Minimum fluorescence ($F_o$) was determined after dark adaptation for at least 15 min. Maximum fluorescence ($F_m$) was obtained by the application of a saturating light pulse (3000 μmol/m$^2$·s) for 0.8 s. The maximum quantum yield of PS II ($F_v/F_m$) was calculated as $(F_m - F_o) F_m^{-1}$ [37,38].

Astaxanthin, which can help cells resist adverse environmental conditions, was used to evaluate the stress condition of cells. The astaxanthin concentration was measured using the spectrophotometric method [39]. Specifically, one milliliter of the sample was mixed with a solution of 5% (*w/v*) KOH in 30% (*v/v*) methanol. The mixture was heated in a 70 °C water bath for 15 min. Then, the mixture was centrifuged, and the supernatant was discarded. The precipitate was extracted with a mixture of DMSO and acetic acid (1.23% of acetic acid) and bathed in a 70 °C water bath again for 15 min. The mixture was centrifuged to collect the supernatant. Then, 200 μL amounts of the samples were taken and placed on a 96-well transparent plate. The microplate reader was used to determine the absorbance of the combined extracts at 492 nm [40]. Finally, the astaxanthin concentration was calculated according to Equation (4).

$$C \ (\text{mg/L}) = 4.5 \times A_{492} \times A \times V_a/V_b, \tag{4}$$

where $V_a$ is the volume of the extracts, $V_b$ is the volume of the culture sample, C is the astaxanthin concentration, $A_{492}$ is the absorbance of the extract at 492 nm, and A is the correction factor, the value of which is 1.3569.

ROS was measured based on the previous method [41]. One milliliter of the sample was centrifuged (13,000 rpm, 5 min) to collect the cells. The residue was resuspended in phosphate-buffered saline (PBS) and centrifuged again to remove the supernatant. Then, 400 µL of dye working solution (1:1000) was added to the processed sample. The mixtures were stored for 60 min in the dark. After two rounds of centrifugation and resuspension in PBS, 200 µL of the obtained sample was taken and placed in a 96-well black plate, and its fluorescence intensity (EX 488, EM525, Gain100) was determined with a microplate reader [42].

### 2.4. Statistical Analysis

The average values and standard deviations of three replicated samples were calculated, which were expressed as mean $\pm$ SD. In the line charts, a Student's *t*-test was used to evaluate differences between groups. Asterisks indicate statistically significant differences (Student's *t*-test, * $p < 0.05$, ** $p < 0.01$, *** $p < 0.001$). For the bar chart, it was more appropriate to use two-way ANOVA to evaluate differences. Different superscript letters indicate significant differences among treatments (two-way ANOVA with Tukey correction, $p < 0.05$).

## 3. Results

### 3.1. The Effect of Different Light Colors on Algal Growth

Generally, the results (Figure 2a–c) show that with the increase in light intensity, the final cell concentration decreased compared with the control treatment (white light, LL), indicating that light color treatments could not alleviate cellular stress caused by high light intensity ($p > 0.05$). However, it is worth noting that cells always performed better in the red-light treatment out of all of the light intensity regimes, suggesting that cells favor red light more than blue light and white light.

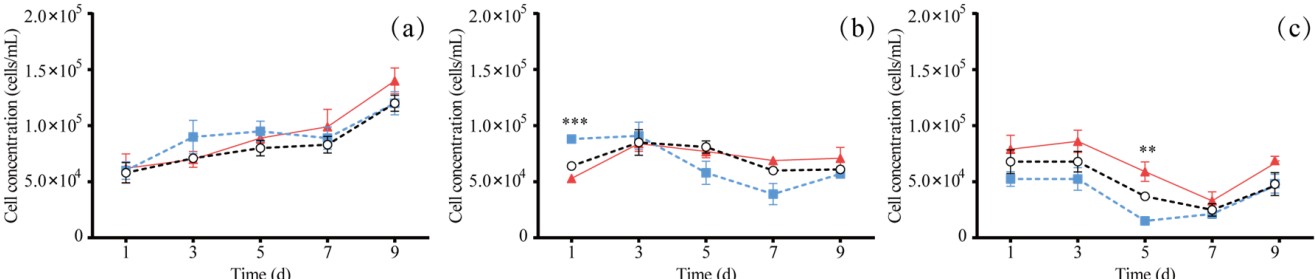

**Figure 2.** Cell concentration curves of *H. lacustris* under light intensities of (**a**) 18 µmol/m$^2$·s, LL; (**b**) 55 µmol/m$^2$·s, ML; (**c**) 110 µmol/m$^2$·s, HL. Treatments are white (open circle), blue (blue box), and red (red triangle) LED lights. Error bars show the standard deviation for three replicates. Asterisks indicate statistically significant differences resulting from Student's *t*-tests: ** $p < 0.01$, *** $p < 0.001$.

Indeed, when comparing the maximum specific growth rate of all of the treatments (Figure 3), it was found that cells grew faster in the red-light treatments. Specifically, the value was 0.34 d$^{-1}$ under the LL condition. Even under the HL condition, the specific growth rate in the red-light treatment was the only growth rate with a value above zero in all of the conditions, which was 0.067 d$^{-1}$.

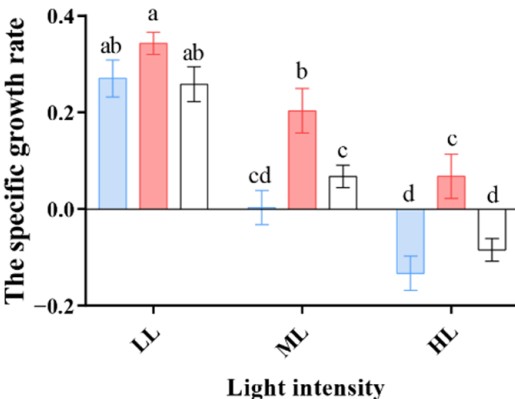

**Figure 3.** The specific growth rate of *H. lacustris* under different light conditions with the blue light (blue bar), red light (red bar), and white light (white bar). Light intensities are 18 μmol/m²·s (LL), 55 μmol/m²·s (ML), and 110 μmol/m²·s (LL). Error bars show the standard deviation for three replicates. Different superscript letters indicate significant differences among treatments (two-way ANOVA with Tukey correction, $p < 0.05$).

The high-level light intensity served as a deterrent to the photosynthetic efficiency of *H. lacustris,* and Fv/Fm declined under unsuitable conditions [38]. From Figure 4a–c, the Fv/Fm value decreased with the increase in light intensity compared with the control treatment (white light, LL). Under the LL intensity (Figure 4a,b), there were no significant differences in the photosynthesis efficiency between red, blue, and white light ($p > 0.05$). However, the photosynthesis efficiency was higher in the red-light treatment under the HL intensity (Figure 4c). In the red-light condition, the Fv/Fm level was higher than in the other two light-color conditions, indicating that red light might be capable of postponing damage to photosystem II.

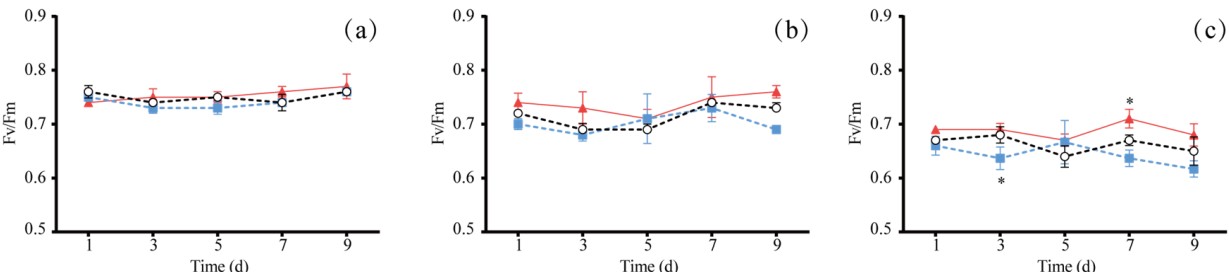

**Figure 4.** Fv/Fm curves of *H. lacustris* under light intensities of (**a**) 18 μmol/m²·s, LL; (**b**) 55 μmol/m²·s, ML; (**c**) 110 μmol/m²·s, HL. Treatments are white (open circle), blue (blue box), and red (red triangle) LED lights. Error bars show the standard deviation for three replicates. Asterisks indicate statistically significant differences resulting from *t*-tests: * $p < 0.05$.

Astaxanthin and ROS concentrations were used to evaluate the stress condition of the cells [43,44]. As Figure 5c–f show, astaxanthin and ROS concentrations increased sharply under the HL intensity. Meanwhile, the LL (Figure 5a,d) and the ML groups (Figure 5b,e) did not display any noticeable changes during the 9-day culture. In the blue light group, under the HL intensity, the astaxanthin concentration reached $5.2 \times 10^{-11}$ g/cell on day 7, and the ROS concentration reached the peak value on day 5 (1.1 unit/cell), which was significantly higher than the concentration achieved in the red-light treatment ($p < 0.05$), indicating that cell damage occurred. Meanwhile, in the red-light condition, the astaxanthin values were lower (the maximum value was $3.0 \times 10^{-11}$ g/cell), and ROS accumulation was postponed until later, reaching 0.98 and 0.96 units/cell after 7 days.

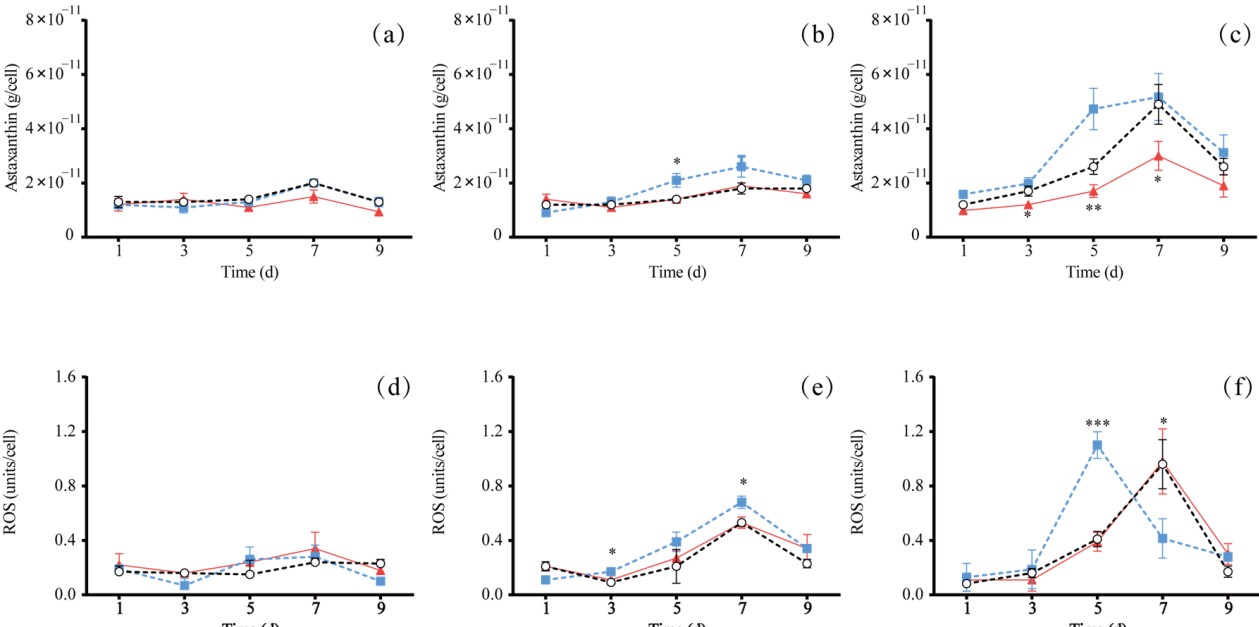

**Figure 5.** Astaxanthin curves of *H. lacustris* under light intensities of (**a**) 18 μmol/m$^2$·s, LL; (**b**) 55 μmol/m$^2$·s, ML; (**c**) 110 μmol/m$^2$·s, HL. ROS curves of *H. lacustris* under light intensities of (**d**) 18 μmol/m$^2$·s, LL; (**e**) 55 μmol/m$^2$·s, ML; (**f**) 110 μmol/m$^2$·s, HL. Treatments are white (open circle), blue (blue box), and red (red triangle) LED lights. Error bars show the standard deviation for three replicates. Asterisks indicate statistically significant differences resulting from Student's *t*-tests: * $p < 0.05$, ** $p < 0.01$, *** $p < 0.001$.

### 3.2. The Effect of Light/Dark Cycles on Algal Growth

Generally, the results (Figure 6a–c) show that the final cell concentration in the dark light treatments increased under the HL intensity compared with the control treatment (white light, LL), indicating that light/dark cycle treatments were capable of alleviating cellular stress caused by high light intensity and resulted in faster growth. The cells performed best in the 2/2 h light/dark cycle treatment under the HL intensity ($p < 0.05$). Among all of the treatments under different light intensity regimes (Figure 6a–c), the highest cell concentration was achieved in the 2/2 h light/dark cycle and 2/2 s light/dark cycle treatment under the ML intensity (Figure 6b, $1.4 \times 10^5$ mL$^{-1}$, $p < 0.5$).

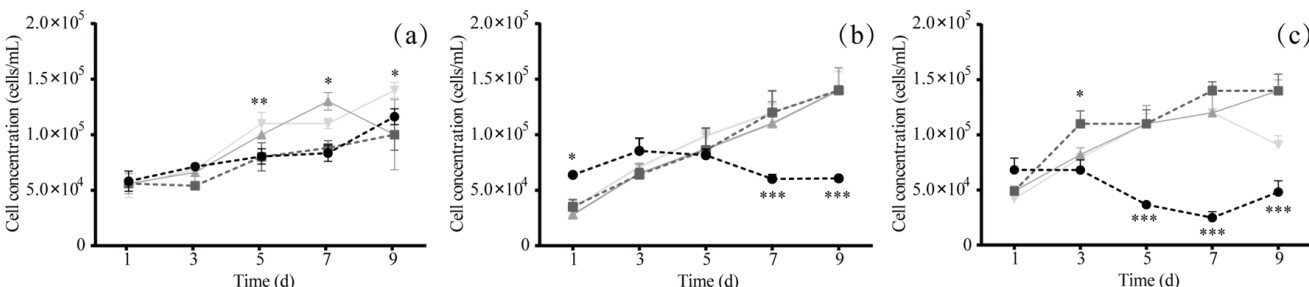

**Figure 6.** Cell concentration curves of H. lacustris under light intensities of (**a**) 18 μmol/m$^2$·s, LL; (**b**) 55 μmol/m$^2$·s, ML; (**c**) 110 μmol/m$^2$·s, HL. Continuous light (black circle), 2/2 s (black box), 2/2 h (gray triangle), 6/6 h (gray inverted triangle). Error bars show the standard deviation for three replicates. Asterisks indicate statistically significant differences resulting from Student's *t*-tests: * $p < 0.05$, ** $p < 0.01$, *** $p < 0.001$.

When comparing the specific growth rate of all of the treatments (Figure 7), it was found that cells grew the fastest in the 2/2 h light/dark cycle treatment under all three

light intensities. Specifically, the max value was 0.38 d$^{-1}$ under the ML intensity, which was 46% higher than the best value in the low light group (0.26 d$^{-1}$, $p < 0.05$).

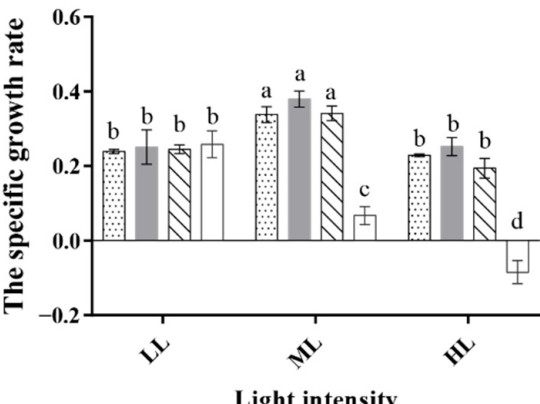

**Figure 7.** The specific growth rate of *H. lacustris* under different light conditions with the light/dark cycle methods. Light intensities are 18 μmol/m$^2$·s (LL), 55 μmol/m$^2$·s (ML), and 110 μmol/m$^2$·s (HL). Continuous light (white bar), 2/2 s (dot bar), 2/2 h (gray bar), and 6/6 h (striped bar). Different superscript letters indicate significant differences among treatments (two-way ANOVA with Tukey correction, $p < 0.05$).

Light/dark cycles might enable remissions of damages to the photosynthetic efficiency of *H. lacustris* from the high light intensity, especially by using a 2/2 h light/dark cycle treatment (Figure 8b,c). It can be seen from Figure 8a that under the LL intensity, there were no significant differences between the various groups ($p > 0.05$). However, as shown in Figure 8b,c, the Fv/Fm of the continuous light group was much lower than the light/dark cycle groups ($p < 0.05$). Additionally, the 2/2 h light/dark cycle could enable the photosynthetic efficiency to remain in a good situation under the HL intensity. These results imply that 2/2 h light/dark cycles might protect photosystem II.

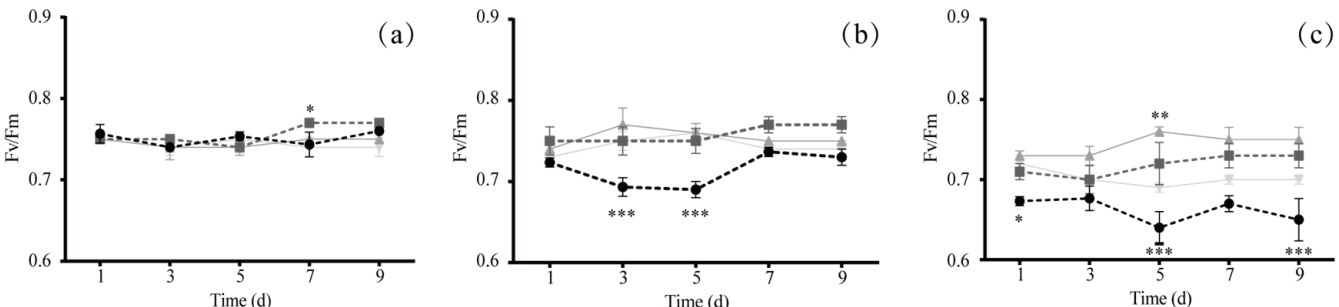

**Figure 8.** Fv/Fm curves of *H. lacustris* under light intensities of (**a**) 18 μmol/m$^2$·s, LL; (**b**) 55 μmol/m$^2$·s, ML; (**c**) 110 μmol/m$^2$·s, HL. Continuous light (black circle), 2/2 s (black box), 2/2 h (gray triangle), and 6/6 h (gray inverted triangle). Error bars show the standard deviation for three replicates. Asterisks indicate statistically significant differences resulting from Student's *t*-tests: * $p < 0.05$, ** $p < 0.01$, *** $p < 0.001$.

The astaxanthin and ROS concentrations remained at low levels under the light/dark cycle, indicating less damage. Comparing Figure 9a–f, it can be seen that the values of the astaxanthin and ROS concentrations could be maintained at a lower level under all light intensities. However, with continuous light treatment, the highest astaxanthin concentration (Figure 9c, $4.94 \times 10^{-11}$ g/cell, $p < 0.05$) and ROS concentration (Figure 9f, 0.964 units per cell, $p < 0.05$) occurred on day 7, indicating cell damage.

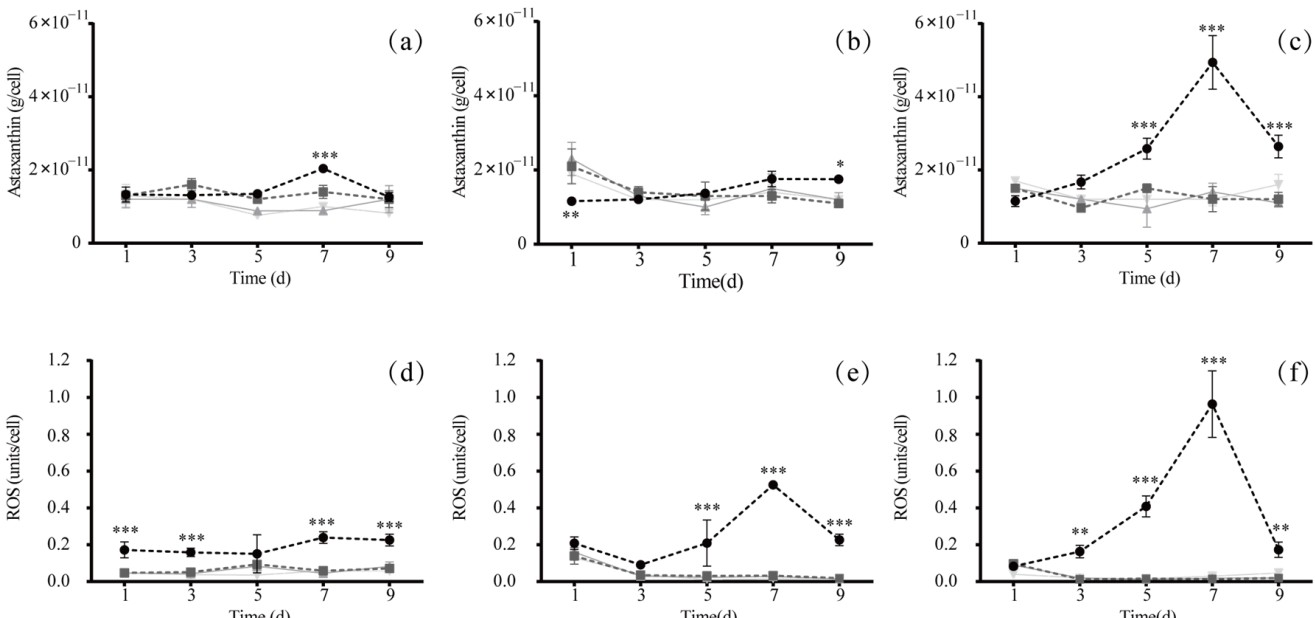

**Figure 9.** Astaxanthin curves of *H. lacustris* under light intensities of (**a**) 18 μmol/m²·s, LL; (**b**) 55 μmol/m²·s, ML; (**c**) 110 μmol/m²·s, HL. ROS curves of *H. lacustris* under light intensities of (**d**) 18 μmol/m²·s, LL; (**e**) 55 μmol/m²·s, ML; (**f**) 110 μmol/m²·s, HL. Continuous light (black circle), 2/2 s (black box), 2/2 h (gray triangle), and 6/6 h (gray inverted triangle). Error bars show the standard deviation for three replicates. Asterisks indicate statistically significant differences resulting from Student's *t*-tests: * $p < 0.05$, ** $p < 0.01$, *** $p < 0.001$.

### 3.3. The Effect of 2/2 h Light/Dark Cycles with Red Light on Algal Growth

From the previous experiments, we found that treatments with red light and a 2/2 h light/dark cycle had a better effect on *H. lacustris* growth under higher light intensities. Therefore, in this section, we used a 2/2 h light/dark cycle with red light to exploit the growth-promoting benefit of increased light intensity while avoiding growth arrestments in *H. lacustris*. According to Figure 10a, treatment with a 2/2 h light/dark cycle with red light could significantly increase the cell concentration under the ML intensity (Red-2h-ML, $p < 0.05$). Additionally, as shown in Figure 10e, the treatment could enable the specific growth rate to reach 0.26 d$^{-1}$ on day 13, which was 66.5% higher than that in the control group (White-24h-LL, 0.156 d$^{-1}$, $p < 0.05$). The Fv/Fm value was stable at 0.74 in the Red-2h-LL group, indicating that photosystem II was protected (Figure 10b). In comparison, the value of Fv/Fm was shown to decrease to 0.7 (Figure 10e) in the control group (White-24h-LL). ROS and astaxanthin concentrations were reduced in the Red-2h-ML and Red-2h-LL groups compared with the control group (White-24h-LL, Figure 10c,d). It was apparent that the damage in cells was alleviated. Thus, treatment with a 2/2 h light/dark cycle with red light could avoid growth arrestment in *H. lacustris*. Additionally, it could exploit the growth-promoting benefit of a higher light intensity of 55 μmol/m²·s (ML), making the specific growth rate 66.5% higher than the control group.

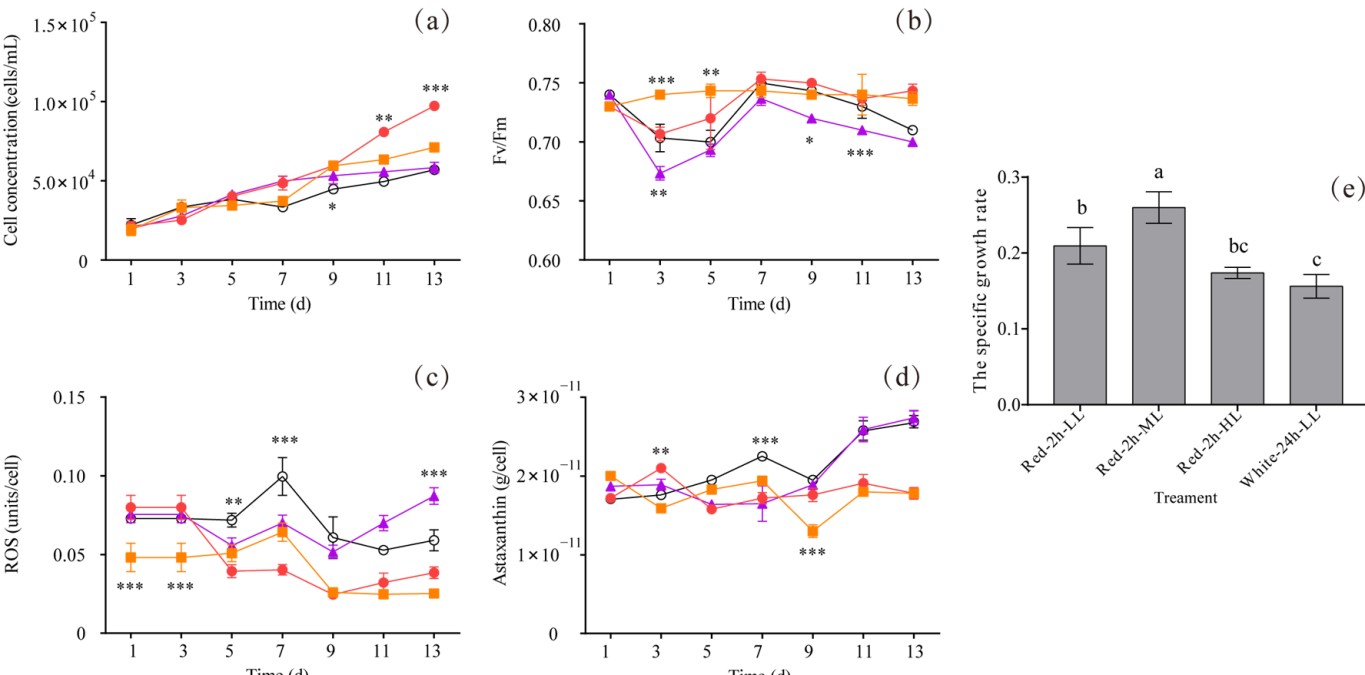

**Figure 10.** Cell concentration curves (**a**), Fv/Fm curves (**b**), ROS curves (**c**), astaxanthin curves (**d**), and the specific growth rate (**e**) of *H. lacustris* under combination treatments. The 2/2 h light/dark cycle with red light under 18 μmol/m²·s (Red-2h-LL, orange box), the 2/2 h light/dark cycle with red light under the μmol/m²·s (Red-2h-ML, red circle), the 2/2 h light/dark cycle with red light under 110 μmol/m²·s (Red-2h-HL, purple triangle). Control group: continuous white light under 18 μmol/m²·s (White-24h-LL, open circle). Error bars show the standard deviation for three replicates. In the line charts, asterisks indicate statistically significant differences (Student's *t*-test, * $p < 0.05$, ** $p < 0.01$, *** $p < 0.001$). In the bar chart, different superscript letters indicate significant differences among treatments (two-way ANOVA with Tukey correction, $p < 0.05$).

## 4. Discussion

To achieve higher biomass, an increase in light intensity is vital. Based on Equation (1), the growth rate of *H. lacustris* could be enhanced by the increasing light intensity. However, *H. lacustris* cells are very sensitive to high light intensity, i.e., cells stop division under 60 μmol/m²·s, which is favored by other algal species [16]. In an attempt to alleviate the adverse effects of high light intensity, different light profiles were used. The results show that a light/dark cycle with red light under the ML intensity could increase the biomass. Speculative reasoning for these results is provided below.

In the first experiment, the results showed that *H. lacustris* growth could benefit most from red light out of all of the light colors. It was a suitable light source for *H. lacustris* growth under higher light intensities. It could improve the PSII efficiency (as shown by the higher Fv/Fm values, Figure 4c) and lessen astaxanthin and ROS accumulation (Figure 5c,f), giving cells a proper light profile and partly preventing cells from being damaged. According to quantum theory, light energy is delivered in photons. The energy of a single light photon is the product of its frequency and Planck's constant, $E = h\nu$ ($h = 6.626 \times 10^{-34}$ J s). As energy is inversely related to wavelength, a blue-light photon (about 400–500 nm) is more energetic than a red-light photon (about 600–700 nm) [45]. High photon energy can cause cell damage [46]. The light energy passes to the photosystem reaction center through the antenna [45]. The energy of blue photons may be higher than the requisition for photosynthesis [47]. Therefore, red light might be more suitable for growth, as it was shown to make ROS and astaxanthin concentrations lower than those achieved with white and blue light. However, compared with the 2/2 h light/dark cycle, the effect was limited.

The first experiment also showed that the 2/2 h light/dark cycle with white light could benefit *H. lacustris* growth. The 2/2 h light/dark cycle treatment achieved a high specific growth rate under the ML intensity ($0.38 \text{ d}^{-1}$, White-2h-ML), which was 46% higher than the control ($0.26 \text{ d}^{-1}$, $p < 0.05$). The 2/2 h light/dark cycle could achieve a high specific growth rate of *H. lacustris* under the ML intensity. In short, light/dark cycle treatments reduced the concentrations of ROS and astaxanthin. This could protect PSII, maintaining Fv/Fm at a normal level. The excessive oxygen produced by light reactions under high light intensities can be used for respiration in the dark period [48]. However, when photon energy accumulates, the pathway could become abnormal [49]. Under continuous light, when the light intensity is increased, excessive oxygen accumulation might be strengthened. As a result, the oxygen will combine with chlorophyll P680 triplet to produce ROS. The oxygen consumption could be increased by utilizing a dark period between two light periods. This could give cells a chance to utilize excessive oxygen generated by light reactions [45]. This might reduce the putative unwanted high ROS and prevent *H. lacustris* from photodamage.

In the second experiment, the 2/2 h light/dark cycle with red light under the ML intensity (Red-2h-ML) showed the optimal growth rate. The combined treatment could achieve a specific growth rate 66.5% higher than the control treatment (White-24h-LL). The level of growth in the second experiment was 44.6% higher than in the first experiment (White-2h-ML, 46% higher than the control). Under the Red-2h-ML treatment, ROS and astaxanthin concentrations were lower, and Fv/Fm was maintained at a higher degree. Cell damage might be alleviated by simultaneously reducing photon energy and promoting oxygen utilization, which could enhance algal growth.

An unexpected phenomenon was observed in the first experiment: the 2/2 h light/dark cycle achieved better results than the 2/2 s light/dark cycle. The potential reasoning for this is as follows: Light attenuation took place during the light/dark switching process. In the 2/2 s condition, light/dark switching was more frequent. Therefore, the average light intensity might have been lower than that in the 2/2 h light/dark cycle, resulting in a lower specific growth rate [50].

Above all, red light could limit the energy of photons, which would lessen the damage to cells; meanwhile, the 2/2 h light/dark cycle could promote oxygen consumption, making ROS levels lower than those in the control treatment. Therefore, the 2/2 h red-light treatment could make *H. lacustris* utilize higher light intensity and grow faster.

## 5. Conclusions

Here, a new light profile strategy to promote *H. lacustris* growth was presented. In the first experiment, light color treatments did not significantly increase the specific growth rate under a higher light intensity. However, compared with other light colors, the photodamage to *H. lacustris* could be slightly mitigated by red light. Under the HL intensity, the specific growth rate of the red-light treatment was the only rate that had a value above zero. Additionally, in the 2/2 h white light treatment, the specific growth rate was 46% higher than that in the control treatment ($18 \text{ µmol/m}^2\cdot\text{s}$, white light) under the light intensity of $55 \text{ µmol/m}^2\cdot\text{s}$, which was the highest among all of the light/dark treatments. Therefore, in the second experiment, a 2/2 h light/dark cycle with red light was used to promote cell growth. The results show that the specific growth rate was 66.5% higher than that in the control treatment ($18 \text{ µmol/m}^2\cdot\text{s}$, white light) under the light intensity of $55 \text{ µmol/m}^2\cdot\text{s}$. These results can help us to optimize light culture schemes. However, the related molecular mechanisms need to be studied in further detail.

**Author Contributions:** Conceptualization, Z.C. and K.Y.; methodology, K.Y. and L.L.; formal analysis, L.L. and K.Y.; investigation, L.L., K.Y. and K.W.; writing—original draft preparation, L.L.; writing—review and editing, S.T., J.Z. and K.W.; visualization, L.L.; supervision, Z.C. and J.Z.; funding acquisition, Z.C. and J.Z. All authors have read and agreed to the published version of the manuscript.

**Funding:** This work was funded by the S&T Projects of Shenzhen Science and Technology Inno-vation Committee (JCYJ20200109142822787, JCYJ20200109142818589, RCJC20200714114433069), as well as the Project of Shenzhen Municipal Bureau of Planning and Natural Resources (No. 2021-735-927#).

**Institutional Review Board Statement:** Not applicable.

**Informed Consent Statement:** Not applicable.

**Data Availability Statement:** Not applicable.

**Acknowledgments:** I would like to thank Zhonghua Cai, Kezhen Ying, Kebi Wu, Si Tang, Jin Zhou, and my family for supporting and helping me to complete this research.

**Conflicts of Interest:** The authors declare no conflict of interest.

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
