# Peer review of "Promoting the Growth of Haematococcus lacustris under High Light Intensity through the Combination of Light/Dark Cycle and Light Color"

_jmse, doi:10.3390/jmse10070839_

Round 1

Reviewer 1 Report

In this article, a strategy to promote Haematococcus lacustris growth is tested. The results indicate that 2h/2h Light/dark and Red Light are the best combination.

The paper is not well written. There is a lot of repetitive and redundant information, especially in Results.

English language should be improved to make this article appropriate for publication.

It is important to present all the images included in the text. If they were not presented they should not be there.

The cited references are mostly old publications (from more than 5 years ago).  Should be updated with more recent references

In Introduction:

Line 26: The first time that the microalgal species name appears in the text (excluding the abstract) should be the complete species name, Haematococcus lacustris.

Line 55: "Light colors absorbed by microalgae differ on the species" Please explain the meaning of this phrase.

Line 73-74: "But how H. lacustris reflect by using light/dark cycle treatments is still unknown." The microalga reflects what? Please explain the meaning of this phrase.

Line 74-78: A lot of information is repeated in both sentences. It would be better to rephrase.

In Materials and Methods:

Line 86: "the flasks were shaken manually six a day" It should be "six times a day"

In this section, it is not advisable to use the first person.

Line 120: In these assays, the cell concentration used was lower. Why was the concentration used in this group of experiments different from the others? 

Line 147: What is "FI"? It must first be defined in the text before using the acronym.

Line 174: "PBS" must first be defined in the text before using the acronym.

In Results:

Line 269: "which was 43% of the best." I don't follow. What do you mean by this statement?

Line 281-282: "But it could make cells utilize the high light intensity and increase the specific growth rate." Explain this sentence better, it is very confusing.

Fig. 9, line 296: Correct the image caption, the intensity values do not appear

Line 301: "Higher photosynthetic efficiencies occurred when microalgae were subjected to light/dark cycles than the constant light. It could protect photosystem II." How did you measure/quantify the photosynthetic efficiencies? The second phrase doesn't make sense in the form in which it is written. 

Line 331: Why did you decide that the best light color was red light and the best light/dark cycle was 2h/2h? Was the decision based on growth rates or on the amount of astaxanthin produced?

Line 336, 337, 339, 340: What is "ML", "HL" and "LL"? It must first be defined in the text before using the acronym.

In Discussion:

Line 419, 420: "Since energy is inversely related to wavelength, a photon of the blue light (about 419 400-500 nm) is more energetic than the red light (around 600-700 nm)" Line 426: "Red light photons are less energetic than blue light photons" Repeated information

Line 428, 429: "In the combined treatment, the red light made ROS and astaxanthin concentrations lower than the white and the blue light treatments" I don't understand the main goal of this experiment, is to grow microalgae or produce astaxanthin? If it is only to grow the microalgae why mention the astaxanthin concentrations? 

Author Response

On behalf of my co-authors, we thank you very much for allowing us to revise our manuscript, we appreciate your constructive comments and suggestions on our manuscript entitled "Promoting the Growth of Haematococcus lacustris by the Light/dark cycle under the Red Light".(ID: jmse-1727804) We have studied the reviewer's comments carefully. The manuscript has been revised and the work is finished. The response and revised manuscript have been uploaded. We would like to thank you again for taking the time to review our manuscript.

Reviewer 2 Report

The peer-reviewed article jmse-1727804 is the result of a successfully performed significant amount of cultural physiological and biophysical study of a promising biotechnological species of algae. All experimental indicators are substantiated, and the obtained results are statistically processed and confirmed. The whole set of materials and the article in general deserve publication. However, if the conducted experiments and the obtained results are sound and deserving of approval, the choice of the section "Marine Ecology" for the submission of the article is unsuccessful due to the object of research. The studied species is freshwater and its cultivation was also carried out under conditions and nutrient medium for freshwater organisms.
It is logical that in such circumstances it is necessary to provide arguments or explanations (authors or editors, with the approval of the article before publication) on the work that the object of study does not correspond to the format of the section.The peer-reviewed article jmse-1727804 is the result of a successfully performed significant amount of cultural physiological and biophysical study of a promising biotechnological species of algae. All experimental indicators are substantiated, and the obtained results are statistically processed and confirmed. The whole set of materials and the article in general deserve publication. However, if the conducted experiments and the obtained results are sound and deserving of approval, the choice of the section "Marine Ecology" for the submission of the article is unsuccessful due to the object of research. The studied species is freshwater and its cultivation was also carried out under conditions and nutrient medium for freshwater organisms.
It is logical that in such circumstances it is necessary to provide arguments or explanations (authors or editors, with the approval of the article before publication) on the work that the object of study does not correspond to the format of the section.The peer-reviewed article jmse-1727804 is the result of a successfully performed significant amount of cultural physiological and biophysical study of a promising biotechnological species of algae. All experimental indicators are substantiated, and the obtained results are statistically processed and confirmed. The whole set of materials and the article in general deserve publication. However, if the conducted experiments and the obtained results are sound and deserving of approval, the choice of the section "Marine Ecology" for the submission of the article is unsuccessful due to the object of research. The studied species is freshwater and its cultivation was also carried out under conditions and nutrient medium for freshwater organisms.
It is logical that in such circumstances it is necessary to provide arguments or explanations (authors or editors, with the approval of the article before publication) on the work that the object of study does not correspond to the format of the section.

Author Response

(The authors gave the same response as above.)

Reviewer 3 Report

This manuscript presented growth and astaxanthin accumulation in Haematococcus with different regime of light, particularly regarding duration and spectra. However, overall design of experiments was flawed and did not achieve any valuable data in terms of biology and biotechnology.

Overall astaxanthin was produced most in the control (white continuous HL light) as shown in Figures 7c, 11c, and 12d. Then, what's the point performing all these experiments? Under these conditions, cell may not be happy, but they produce most astaxanthin, which is the main target molecule from Haematococcus (as Authors claimed).

One thing authors can do is assessing astaxanthin yield rather than astaxanthin contents per cell.

In addition, authors assessed photosynthetic efficiency with Fv/Fm; however, NPQ should also be measured, particularly if ROS is the problem, which is usually resulted from antennal overload. Most microalgae are good with NPQ, so they are happy with high light. It seems likely that Haematococcus is poor with NPQ, which would be interesting to see.

Statistical significance? Authors presented a lot of data in graphs; however, statistical significance was not, leaving audience guessing about differences. In some cases, authors claimed differences (highest, lowest, etc.) even when individual graphs overlap with error bars.

The way authors chose red light for further experiment leaves questions. In Figure 4 blue light showed better growth under certain conditions, particularly with low light. However, they used red light only with low light for LD cycles in Figure 12. Blue light should also be included for these experiments.

Overall growth curves fluctuate so much that integrity of data is in question.

More specific issues follow:

Introduction:

- line 55: light colors > light spectra?

- line 59: Spirulina is not green algae.

- line 60: "But for Nannochloropsis sp, the blue light is the best" – Best for what?

- line 67-68: Authors listed old references for the benefits of LD cycle on photosynthesis. It is asked to review latest papers for specific benefits of LD cycle, and then associate this to the current manuscript.

In addition, authors cited cyanobacterial papers. It is also asked to distinguish bacterial differences including antennas, pigments, and photoreceptors, compared to eukaryotic algae.

Methods:

- line 96: "In light color treatments, the light period lasted 24 hours each day." Not clear of the meaning.

- line 117: It is not clear how the red light was administered. Was it on all the times regardless of LD cycle? Or, was it off during the dark period? Or, maybe, the red light was used in LD cycle. It is confusing because authors are saying "LD cycle and red light," however, based on Figure 3, it seems like that the red light was used in LD cycle. Please make this clearer.

- line 144: fluorescence value (OD600)?

- Authors determined cell concentration from data of OD600 using their standard curve, which may present errors, particularly for biphasic growth (as in Haematococcus), where cell division and cell size growth occurs. I am wondering if they took cell size mistakenly for cell number, or vice versa. I wish they just took OD600 data directly for growth curve, which would represent biomass. Biomass is more important biotechnological criteria, related to the yield of molecules (astaxanthin in this case) derived from biomass.

Results:

- Figure legends include graphics for markers, which causes formatting problems. These should be replaced with text description, such as open circle, blue box, red triangle, etc.

- Figures 1-3 should be reorganized and combined into one figure, which may be in a supplementary figure.

- Figure 4, significance (e.g., student's t test) should be assesses to describe differences.

- Figure 5: In my understanding, specific growth rate should be measured during the exponential phase, and in Figure 4a, blue light showed highest SGR. SGR should be measured strictly based on the definition.

For other growth curves, for HL, all samples (W, R, B) showed decreased cell concentration at the end. I do not see how authors calculated SGR.

- Figure 7: Astaxanthin and ROS curves were switched? The same for Figure 11?

- Figure 12: Please use different colors for different light intensities. It is hard to distinguish small markers, all in red. Are there any particular reasons to do day 13, while all other experiments were done day 9?

- Figure 13: "Photoinhibition induces the production of ROS" It should be opposite: ROS usually results in photoinhibition.

- English writing should be corrected by professional editors to make sentences clearer and to avoid confusing readers. Some examples:

line 10: proposed should be accepted

line 17: "Nonetheless, light color treatments could not increase the growth apparently, whereas the red light could only slightly mitigate the photodamages." This sentence is confusing and may be incorrect. Firstly, "light color treatments could not increase the growth apparently" – Explain why not? And then, red light can have some effect?

line 18: thus should be therefore

line 19: "growth performance" Would it be growth pattern? Or just growth. In addition, the sentence should be re-written to explain experimental scheme better.

line 20: Specify the control group.

These confusing/incorrect sentences could be found in the later parts, which should be corrected.

Author Response

(The authors gave the same response as above.)

Round 2

Reviewer 1 Report

Thank you for the answers. The article is much better after the changes made and it is easier to understand the purpose of the study.
Most of the issues have been cleared with the exception of:

- Figure 1 is still not presented in the text.

Author Response

Thank you very much for your comments, we learned a lot during the revision process. We appreciate your suggestions. Thank you for helping us make our manuscript much better than before.

Reviewer 3 Report

This manuscript appears to describe growth promotion by red light, for which authors excluded biotechnological merits in terms of astaxanthin and biomass yields. From biological perspectives, data were not presented enough to support growth promotion by red light in different irradiation scheme. The following should be addressed for publication in JMSE.

- Title: "Light/dark cycle under the Red Light" This sounds like that the red light is irradiated from the top, and additional light is administered in LD cycle. Is this true? However, based on Figure 1C, red light was administered in the LD cycle, which should be "Light/Dark cycle with the red light."

- Abstract:

- Line 17: "cells favored the red light the most in all light colors" Not clear what this meant. Would it be "cells favored the red light the most among all colors?"

Point 5 about statistical significance: Authors described t test for this, but not indicated on any data. If P < 0.05, asterisks should be marked on actual graphs, particularly on specific growth rates (figures 3, 7 and 10e), since authors are focused on “promoting the growth of Haematococcus” as in the title. Authors did not show any significant difference on these key data, so they are not supposed to claim any improvement of growth.

Point 17 about specific growth rate: I understand that the cited reference (Yu et al 2017 and many others, also including me) have used the same equation. However, all these are for microalgae showing typical growth curves as shown for Scenedesmus and Chlorella. On the other hand, Haematococcus in the manuscript hardly show such growth pattern, and cells showed fluctuation to make things worse. In this case, t0 and t1 should not be just the start and the end of experiments, respectively. Please refer to other Haematococcus papers, and follow how they determined SGR.

- Figure 3: The SGR data is supposed to be originated from Figure 2; however, its data are grouped based on color, while Figure 2 is grouped based on light intensity. This is confusing, and again authors did not show any statistically significant difference (P<0.05), so all treatments resulted in the same (or insignificant) growth pattern. In particular, with LL, all lights (white, blue, red) were the same and then why red was chosen for further experiments?

- Figure 7: LD cycles of 2s, 2h, and 6h were similar, and authors did not indicate any statistic differences between them. Question would be why authors thought 2h/2h cycle was the best?

- Figure 10: Control for these experiments should also include white-2h-ML to show "promoted growth" by red light in LD cycle. In addition, based on my understanding, Figure 7 showed continuous white light (LL) was the same as (or even higher than) LL of different light cycles, but in Figure 10, White-24h-LL (I think this is the continuous light) appeared to be the lowest. Again, author did not indicate statistical significance. Please clarify the discrepancy.

In addition, Figure 10 showed data up to day 13, while all other data up to day 9. Explain why. This also brings another question about growth phase of Haematococcus. Please define different growth phases specifically in terms of culture time.

- Figure 11. I am not an expert in photosynthetic machinery, but understand that P680 triplet should precede QA reduction, then producing ROS. In addition, authors indicated "high energy photon." Is there low energy photon? It should also be pointed out that ROS may not be produced by this mechanism in Haematococcus, simply because none of the intermediates were analyzed in this manuscript. High light intensity (110) used in this manuscript is not high light for other microalgae (they are happy with this light), and thus Haematococcus may have different ways for such high light sensitivity. Overall, this model is not supported by any data presented in this manuscript, which can be removed.

Author Response

Thank you very much for your patience, we have learned a lot during the revision process. We appreciate your serious attitude, and we have revised the paper according to your comments. Thank you for helping us make our manuscript much better than before.

Round 3

Reviewer 3 Report

Thank you for your efforts and patience with my reviews.